# Ant Colony Optimization with Warm-Up

**Mattia Neroni** 

Department of Engineering and Architecture, University of Parma, 43124 Parma, Italy; mattia.neroni@unipr.it

**Abstract:** The Ant Colony Optimization (ACO) is a probabilistic technique inspired by the behavior of ants for solving computational problems that may be reduced to finding the best path through a graph. Some species of ants deposit pheromone on the ground to mark some favorable paths that should be used by other members of the colony. Ant colony optimization implements a similar mechanism for solving optimization problems. In this paper a warm-up procedure for the ACO is proposed. During the warm-up, the pheromone matrix is initialized to provide an efficient new starting point for the algorithm, so that it can obtain the same (or better) results with fewer iterations. The warm-up is based exclusively on the graph, which, in most applications, is given and does not need to be recalculated every time before executing the algorithm. In this way, it can be made only once, and it speeds up the algorithm every time it is used from then on. The proposed solution is validated on a set of traveling salesman problem instances, and in the simulation of a real industrial application for the routing of pickers in a manual warehouse. During the validation, it is compared with other ACO adopting a pheromone initialization technique, and the results show that, in most cases, the adoption of the proposed warm-up allows the ACO to obtain the same or better results with fewer iterations.

**Keywords:** heuristics; traveling salesman problem; TSP; ant colony optimization; metaheuristic; warm-up

## 1. Introduction

The Ant Colony Optimization (ACO) is a combinatorial optimization technique inspired by the behaviour of some species of ants. Broadly, when an ant must choose one route instead of the other, he/she looks at the quantity of pheromone left by other members of the colony. A higher level of pheromone means a better route, usually because it is shorter if compared to the others. This curious behavior inspired the creation of a probabilistic technique of operational research for solving computational problems, which can be reduced to finding the best path through a graph. The first version was proposed by [1], and it was originally called Ant System. Since then, many versions and different applications of the ACO were studied, and the algorithm is nowadays known to be a well-established and efficient approach for many practical problems, primarily the well-known traveling salesman problem (TSP) [2]. Consequently, an improvement in the ACO would lead to great benefits in many industrial and nonindustrial fields. Being the ACO a metaheuristic algorithm, most of the problems approached with it are strictly time-critical. Usually, they are NP-hard problems, in which the global optimum is refused a priori to seek a reasonably good suboptimal solution. However, the ACO, like all the evolutionary algorithms, needs many iterations to converge to a good solution, and, in the case of large-size problems, this process can be very time-consuming [3]. For this reason, the implementation of the ACO for solving large-size problems in real-time (i.e., a few seconds or even less) might be problematic. This is the first open problem highlighted also by [4], and this is because the adoption of a technique able to speed up the ACO may be very useful.

In this paper, a warm-up procedure to reduce the number of iterations required by the ACO to converge to a good solution is proposed. The success of the ACO is essentially based on a variable called the pheromone matrix. The pheromone matrix registers the

current quantity of pheromone on each edge of the graph, and it, therefore, determines the probability to include each specific edge in a newly generated solution. In the classic version of the ACO, every time the algorithm is executed, all the elements of the pheromone matrix are set equal to a starting (generally low) value. Then, as the computed iterations increase, the pheromone on the most promising paths is increased and that on the less convenient paths is reduced. Although, in most real implementations, the graph of nodes is given and is the same in every execution. Furthermore, it was verified that, given a graph, in many cases, each time the ACO was executed, after a certain number of iterations, the pheromone matrix was always very similar. The warm-up procedure proposed in this paper aims to carry out a fine-tuning of the pheromone matrix on a specific graph, so that, every time the ACO is executed, it starts from an already weighted graph, where the promising paths were highlighted with a high level of pheromone and the bad paths excluded a priori. This process is supposed to reduce the number of iterations that the ACO requires to converge every time it is executed. The remainder of this paper is organized as follows. Firstly, a brief overview of the scientific contributions to ACO is presented in Section 2. Then, the warm-up procedure proposed in this paper is described in Section 2. The computational experiments are shown in Section 4, where the ACO with warm-up is compared to the classic ACO, and two other ACO versions that carry out an initialization of the pheromone matrix. Finally, the conclusions are presented in Section 5.

## 2. Literature Review

The Ant Colony Optimization (ACO) was introduced by [1] as a novel nature-inspired metaheuristic for the solution of hard combinatorial optimization problems. ACO belongs to the class of metaheuristics, which are approximate algorithms used to obtain good enough solutions to NP-hard problems in a reasonable amount of time. When searching for food, some species of ants initially explore the area surrounding the nest randomly. As soon as an ant finds a food source, it evaluates the quantity and the quality of the food. On the way back, the ant deposits a chemical pheromone trail on the ground. The quantity of pheromone deposited depends on the quantity and quality of the food and guides other ants to the food source. As shown by [5], the communication via pheromone between the ants enables them to find the shortest paths between their nest and food sources, and the same consideration also applies in ant colony optimization algorithms for solving combinatorial optimization problems. Even if the first proof-of-concept application for the ACO was a traveling salesman problem (TSP), up to now the above algorithm was applied to many combinatorial optimization problems. For instance, it was applied to assignment problems [6–8], routing problems [9–11], scheduling problems [12,13]. Less known but equally efficient applications concern the resource-constrained project scheduling problem [14,15], flow shop scheduling [16], sequential ordering problem [17], and open shop scheduling problem [18]. The scientific community also proposed many applications for nonindustrial environments such as solutions for DNA sequencing or web page ranking [19]. The various variants of the ACO generally differ from each other in the pheromone update rules. In particular, most applications belong to one of these two categories: the iteration-best-update or the best-so-far-update. Basically, in the first case, the update of pheromone takes place at every iteration, while in the second case it takes place only when a new best solution is found, introducing in this way a much stronger bias towards the good solutions found. The most successful ACO variants are the Ant Colony System [20] and the Min-Max Ant System [21], which also are the most used in practice. Since this claims to be just a brief overview of the key points concerning ACO, for a deeper analysis of the scientific contributions on this algorithm the literature reviews by [4,22] are suggested.

### 3. Warm-Up

*3.1. General Considerations*

In most applications where the ACO is used or might be used, the graph of nodes that characterizes the problem is given and constant. Consequently, even the matrix of the costs associated with the edges is constant. This is well-known by the practitioners and affirmed by many scientific publications: see for example [4,23,24], which, indeed, takes the matrix of the costs as given. For instance, in classic traveling salesman problems for vehicle routing or picker routing in manual warehouses, the nodes represent the locations to visit and the costs associated with the edges represent the distance between the two connected nodes. Hence, the matrix of distances does not change until the roads network changes (in case of vehicle routing) or the warehouse layout changes (in case of picker routing). As matter of fact, in almost all the papers that treat these topics, the matrix of distances is defined only once using an exhaustive algorithm such as Floyd-Warshall to find the shortest path between all the nodes of the graph (see for instance [10]). Hence, when the graph and the matrix of costs are formalized, a warm-up may also be carried out. The warm-up allows a tuning of the pheromone matrix used by the ACO, and, in this way, every time the ACO is executed from then on, the number of iterations it needs to converge to a good solution is reduced, and, consequently, its computational time is reduced. The aim of the warm-up is therefore to highlight a priori the most promising paths, as well as excluding a priori the worst ones. All these aspects were already affirmed and well-described also by other scientific contributions focused on the initialization of the pheromone matrix (see for instance [25,26]).

*3.2. The Notation Used*

In the remainder of this section, for describing the proposed procedure, the following notation is used.

- $m = 1, \ldots, M$ are the iterations of the warm-up process;
- $i, j = 1, \ldots, N$ are the nodes of the graph;
- $C$ is the matrix of costs, where each element $c_{i,j}$ is the cost associated to the edge $(i, j)$;
- $T$ is the pheromone matrix of the ACO, where each element $\tau_{i,j}$ is the pheromone on the edge $(i, j)$;
- $P(m)$ is the matrix of probabilities in iteration $m$, where each element $p_{i,j}(m)$ is the probability to increase the pheromone on the edge $(i, j)$ during the iteration $m$;
- $U(m)$ is the matrix of updates in iteration $m$, where each element $u_{i,j}(m)$ says how much the pheromone on the edge $(i, j)$ is supposed to be increased during the iteration $m$;
- $\alpha, \beta, \rho, Q, \tau_0$ are classic and well-known parameters of the ACO [1]: $\alpha$ and $\beta$ define the probability to select a specific edge according to the pheromone on it, $\rho \in [0, 1]$ is known as evaporation rate and defines the decrease of the pheromone that takes place at each iteration, $Q$ defines the quantity of pheromone laid by the ants at each iteration, and, $\tau_0$ is the starting pheromone on each edge;
- $\rho_{wu}$ is a variant of $\rho$, with a different value used during the warm-up;
- $I_A$ is the identity matrix of size $A x A$;
- $\circ$ is the Hadamard product.

*3.3. The Procedure*

The warm-up emulates the update of the pheromone that, in classic ACO, is made during the first iterations of the algorithm, i.e., those generally aimed to explore the graph. The procedure is iterative and relatively easy. First of all, the pheromone matrix $T$ is initialized, setting each element $\tau_{i,j} = \tau_0$ ( $\forall \tau_{i,j} \in \{1, \ldots, N\} | i \neq j$) and $\tau_{i,j} = 0$ ( $\forall \tau_{i,j} \in \{1, \ldots, N\} | i = j$). Similarly, to avoid divisions by zero, all the elements on the diagonal of the matrix of costs

are made equal to 1 ($C = C + I_N$). At each iteration $m$, the probability matrix $P(m)$ is built by calculating each of its elements as in the following equation.

$$p_{i,j} = \frac{(\tau_{i,j})^\alpha \cdot (\frac{1}{c_{i,j}})^\beta}{\sum_{j=0}^{N} (\tau_{i,j})^\alpha \cdot (\frac{1}{c_{i,j}})^\beta} \quad \forall i,j \in 1, \ldots, N \tag{1}$$

Note Equation (1) is the same used by many authors and mentioned by [4] to compute the probability to include the edge $(i, j)$ in the new generated solution at each iteration of the algorithm. Then, the matrix of updates $U(m)$ is calculated as in Equation (2), according to [1].

$$u_{i,j} = \frac{Q}{c_{i,j}} \quad \forall i,j \in 1, \ldots, N \tag{2}$$

Then, the pheromone matrix $T$ is updated according to Equation (3). In particular, the pheromone on each edge is updated according to its cost, and its corresponding value in the matrix of probabilities.

$$T = T + [U(m) \circ P(m)] \tag{3}$$

Finally, the pheromone evaporates as expressed in Equation (4).

$$T = \rho_{wu} \cdot T \tag{4}$$

The process is then repeated until the maximum number of iterations $M$ is reached. In general, it is possible to see how the warm-up emulates exactly the same process that takes place during the iterations of the ACO. However, while during each iteration of the classic ACO only the pheromone on the edges owning to a new generated best solution is increased, in this case, at each iteration, all the edges of the graph see an increase of the pheromone, and this increase is proportional to their attractiveness. This is also a peculiarity of the proposed approach when compared to existing ones in literature (see for instance [25] or [26]), which generally initialize the pheromone simply depending on the cost associated to each edge of the graph. The author is aware that, over the years, several versions of the ACO were proposed by the scientific community, and most of them differ from the others for the formulas adopted to calculate the increase of pheromone [27], the evaporation [9], and the probability to choose an edge instead of the other [28]. On occasion of this study, reference is made to the first version by [1]. As several different versions of the ACO exist, many different versions of this warm-up procedure can be made by doing slight modifications to the formulas.

*3.4. The Parameters Tuning*

Concerning the tuning of parameters, the same setting analyzed and defined as 'optimal' by [1] is used (i.e., $\alpha = 1$, $\beta = 2$, $\rho = 0.9$, $Q = 5$, $\tau_0 = 0.1$). The additional parameters used in the warm-up that need an optimization are the evaporation rate used during the warm-up (i.e., $\rho_{wu}$) and the number of iterations of the warm-up (i.e., $M$). There is no real optimum for these parameters that can be defined a priori; both depend on the size of the problem, its complexity, and the type of connections in the graph. In occasion of this study, to carry out a good setting before the computational experiments described in the next section, three different traveling salesman problem benchmarks are used. Each of these problems consists in the construction of the cheapest Hamiltonian cycle through a set of nodes, and each of them has a different complexity identified by the number of nodes to connect (i.e., 20, 30, 40). Concerning the parameters, three different levels were identified per each of them, i.e., $\rho_{wu} \in \{0.5, 0.9, 1.0\}$ and $M \in \{200, 400, 600\}$, and the ACO with warm-up was tested on each problem using all the possible combinations. Moreover, because of the randomness of the procedure, given a benchmark problem, and a combination of $\rho_{wu}$ and $M$, not just a single execution of the ACO was considered; conversely, the algorithm was executed five times under the same conditions and its average result and standard deviation monitored. The results are reported in Table 1,

where is visible that the best results (those highlighted in greed) are obtained for $\rho_{wu} = 1$ and $M = 400$. As suggested by $\rho_{wu} = 1$, the evaporation should be avoided during the warm up.

**Table 1.** Parameters' tuning.

| $M$ | $\rho_{wu}$ | Problem 1 (# Nodes: 20) | | Problem 2 (# Nodes: 30) | | Problem 3 (# Nodes: 40) | |
|---|---|---|---|---|---|---|---|
| | | Avg. | St.Dev. | Avg. | St.Dev. | Avg. | St.Dev. |
| | 0.5 | 58,526 | 3050 | 61,487 | 5541 | 68,708 | 6035 |
| **200** | 0.9 | 58,765 | 3257 | 65,716 | 6749 | 74,444 | 9170 |
| | 1.0 | 46,455 | 720 | 49,971 | 0 | 55,726 | 948 |
| | 0.5 | 52,711 | 2255 | 65,110 | 5605 | 71,279 | 5114 |
| **400** | 0.9 | 56,299 | 6108 | 67,829 | 4789 | 73,006 | 2915 |
| | 1.0 | 46,017 | 355 | 49,971 | 0 | 55,609 | 1069 |
| | 0.5 | 56,691 | 4502 | 64,197 | 3450 | 73,063 | 5743 |
| **600** | 0.9 | 72,054 | 2056 | 65,191 | 7032 | 81,485 | 4261 |
| | 1.0 | 46,434 | 166 | 50,020 | 306 | 55,800 | 491 |

## 4. Computational Experiments

### 4.1. General Considerations

For validating the efficiency of the proposed warm-up approach, a set of computational experiments is presented in this section. All the experiments carried out are based on the traveling salesman problem (TSP), which, to the author's best knowledge, is also the most frequent and popular application of the ACO. The objective of the algorithm is therefore the definition of a low-cost Hamiltonian cycle: given *(i)* a set of nodes to visit, *(ii)* a set of edges connecting them to each other, and *(iii)* a cost associated to each edge, the algorithm has to define the sequence in which the nodes should be visited that minimizes the total cost of covered edges. Firstly, a set of generic TSP instances is used. In particular, five different graphs are generated, and, on each graph, five different experiments of different complexities are done. Each experiment is taking in consideration a different set of nodes of the graph: the greater is the set of nodes, the higher is the complexity of the problem. Then, to validate the proposed approach in a more realistic context, the simulation of a real industrial case is used. The layout of a manual warehouse for order picking is considered, and the proposed algorithm is used to define the optimal (or almost optimal) paths made by pickers to collect the desired products. No capacity limits are imposed on pickers or aisles, hence the situation is perfectly comparable to a classic TSP, although, the graph is more constrained and has all the characteristics of those used to model warehouses.

### 4.2. The Comparison Algorithms

The proposed ACO with warm-up (ACOWU) is compared to a classic ACO (i.e., without warm-up) having the same parameters setting, and two ACOs using a pheromone initialization technique(i.e., [25,26]).

The first comparison algorithm with pheromone initialization proposed by [26] (hereafter simply referred to as *Dai*) is based on the Minimal Spanning Tree (MST). Given the graph of nodes, once calculated the MST using the well-known Prim's algorithm, and given $\tau_0$ the starting pheromone on nodes, the pheromone on nodes belonging to the MST is set to $\tau_0^{1/\beta}$.

Conversely, the algorithm proposed by [25] (hereafter simply referred to as *Bellaachia*) says to set the pheromone on edge $(i, j)$, namely $\tau_{i,j}$:

$$\tau_{i,j} = \frac{1}{\sum_{z \in N^*}(c_{i,z})} \tag{5}$$

where $N^*$ (i.e., $\subset N$) is the set of nodes, different by $j$, which can be reached by $i$.

### 4.3. Collected Information

The warm-up is made only once on each graph, while at each run of the algorithms three main parameters are controlled: *(i)* the cost of the best solution found, *(ii)* the number of iterations needed to find it, and *(iii)* the computational time. Being all the observed algorithms subject to a certain randomness, to have a better understanding of their reliability, they were all iterated 10 times on each experiment, and the average and standard deviations are therefore reported. For sake of clarity, in all the following tables, the results of the proposed algorithm are written in bold when it outperforms the classic ACO, and highlighted in grey every time it outperforms all the other algorithms.

### 4.4. Results Obtained on Generic TSP Instances

The results obtained on the generic TSP instances are reported in Tables 2–4. In particular, results concerning the cost of the best solution found are reported in Table 2, results concerning the number of iterations needed to find the best solution are reported in Table 3, and computational times are in Table 4. For sake of clarity, the results obtained by the proposed ACOWU are written in bold when it outperformed the classic ACO without warm-up, and highlighted in gray when it outperformed all the other comparison algorithms.

**Table 2.** Results obtained on generic TSP instances in terms of cost.

| G | N | ACO | | ACOWU | | Dai | | Bellaachia | |
|---|---|---|---|---|---|---|---|---|---|
| | | Avg. | St.Dev. | Avg. | St.Dev. | Avg. | St.Dev. | Avg. | St.Dev. |
| 0 | 20 | 4453 | 162 | 4471 | 159 | 4496 | 130 | 5154 | 463 |
| 0 | 30 | 6079 | 513 | **5539** | 118 | 5497 | 230 | 5994 | 134 |
| 0 | 40 | 6915 | 422 | **6468** | 171 | 6782 | 387 | 7421 | 444 |
| 0 | 50 | 7487 | 199 | 7563 | 382 | 7565 | 518 | 8472 | 380 |
| 0 | 60 | 9250 | 392 | **8236** | 276 | 8106 | 166 | 9080 | 884 |
| 1 | 20 | 4276 | 115 | **4062** | 121 | 4146 | 89 | 4993 | 382 |
| 1 | 30 | 5663 | 298 | **5261** | 148 | 5627 | 420 | 6162 | 475 |
| 1 | 40 | 6849 | 589 | **6061** | 373 | 6301 | 226 | 7502 | 794 |
| 1 | 50 | 7550 | 637 | **6940** | 216 | 7414 | 404 | 8152 | 178 |
| 1 | 60 | 9415 | 1278 | **7861** | 408 | 8293 | 828 | 9572 | 479 |
| 2 | 20 | 4693 | 389 | **4494** | 47 | 4503 | 148 | 4964 | 136 |
| 2 | 30 | 5731 | 243 | **5500** | 210 | 5687 | 252 | 6484 | 254 |
| 2 | 40 | 6940 | 594 | **6233** | 269 | 6698 | 408 | 7639 | 585 |
| 2 | 50 | 8853 | 1029 | **7873** | 95 | 9076 | 552 | 9101 | 492 |
| 2 | 60 | 9462 | 348 | **8612** | 271 | 9266 | 801 | 10,457 | 677 |
| 3 | 20 | 3906 | 157 | **3808** | 68 | 3874 | 42 | 4512 | 340 |
| 3 | 30 | 5529 | 241 | **5243** | 141 | 5514 | 185 | 6359 | 459 |
| 3 | 40 | 7097 | 499 | **6562** | 140 | 6955 | 413 | 7914 | 503 |
| 3 | 50 | 7686 | 569 | **7049** | 217 | 7798 | 458 | 8308 | 770 |
| 3 | 60 | 9041 | 470 | **8028** | 150 | 8261 | 628 | 9309 | 585 |
| 4 | 20 | 4729 | 261 | **4407** | 167 | 4510 | 165 | 5206 | 185 |
| 4 | 30 | 5696 | 541 | **5344** | 270 | 5857 | 312 | 6271 | 229 |
| 4 | 40 | 7324 | 404 | **6586** | 266 | 7342 | 344 | 7710 | 331 |
| 4 | 50 | 8092 | 757 | **7400** | 139 | 8113 | 160 | 9145 | 774 |
| 4 | 60 | 9090 | 779 | **8234** | 296 | 8730 | 49 | 9355 | 494 |

As visible in Table 2 and represented in Figure 1, the proposed warm-up allows the ACOWU to outperform all the other algorithms in almost all the experiments. The Dai algorithm also performs better than the classic ACO, but it rarely reach equals the results of the proposed one. The Bellaachia algorithm provides reasonably good results, although it struggle to reach even the ACO. It is reasonable to believe the authors of Bellaachia algorithm focused more on the reduction of iterations needed to converge to a good solution than on the quality of the solution itself.

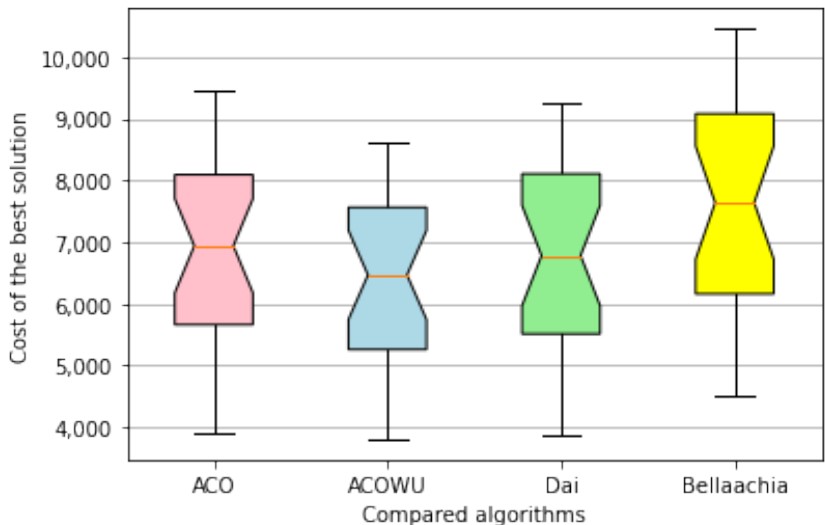

**Figure 1.** Comparison in terms of cost on generic TSP instances.

**Table 3.** Results obtained on generic TSP instances in terms of solutions explored before finding best one.

| G | N | ACO | | ACOWU | | Dai | | Bellaachia | |
|---|---|------|--------|------|--------|------|--------|------|--------|
| | | Avg. | St.Dev. | Avg. | St.Dev. | Avg. | St.Dev. | Avg. | St.Dev. |
| 0 | 20 | 1080 | 944 | 1632 | 627 | 1063 | 737 | 804 | 938 |
| 0 | 30 | 638 | 1143 | 862 | 881 | 700 | 880 | 1157 | 684 |
| 0 | 40 | 992 | 887 | 1497 | 480 | 654 | 491 | 707 | 307 |
| 0 | 50 | 2005 | 549 | **1261** | 648 | 407 | 460 | 977 | 532 |
| 0 | 60 | 1064 | 691 | **647** | 360 | 1666 | 299 | 1078 | 1002 |
| 1 | 20 | 310 | 154 | 919 | 401 | 1100 | 490 | 381 | 373 |
| 1 | 30 | 713 | 428 | 935 | 780 | 728 | 747 | 1092 | 251 |
| 1 | 40 | 672 | 852 | 812 | 666 | 809 | 916 | 1046 | 958 |
| 1 | 50 | 2050 | 1258 | **1112** | 889 | 1581 | 593 | 840 | 497 |
| 1 | 60 | 1186 | 910 | **1081** | 969 | 1274 | 836 | 896 | 687 |
| 2 | 20 | 972 | 715 | **837** | 607 | 883 | 195 | 609 | 387 |
| 2 | 30 | 1598 | 982 | **1083** | 355 | 1321 | 956 | 612 | 370 |
| 2 | 40 | 598 | 277 | 1344 | 896 | 1293 | 794 | 1406 | 728 |
| 2 | 50 | 1072 | 660 | 1262 | 699 | 644 | 549 | 1192 | 953 |
| 2 | 60 | 1542 | 993 | 1613 | 934 | 1275 | 532 | 1394 | 610 |
| 3 | 20 | 1089 | 687 | 1101 | 686 | 987 | 640 | 648 | 447 |
| 3 | 30 | 1509 | 983 | **969** | 596 | 1672 | 881 | 846 | 533 |
| 3 | 40 | 982 | 1044 | 1202 | 977 | 981 | 682 | 973 | 798 |
| 3 | 50 | 1860 | 1034 | **727** | 947 | 866 | 1063 | 885 | 806 |
| 3 | 60 | 570 | 243 | 819 | 369 | 1836 | 900 | 766 | 376 |
| 4 | 20 | 739 | 880 | 1348 | 917 | 1254 | 1067 | 586 | 227 |
| 4 | 30 | 1065 | 607 | 1341 | 912 | 621 | 521 | 907 | 813 |
| 4 | 40 | 1391 | 770 | **458** | 635 | 1575 | 1016 | 911 | 663 |
| 4 | 50 | 1918 | 984 | **868** | 812 | 981 | 634 | 584 | 318 |
| 4 | 60 | 1197 | 1110 | **566** | 357 | 815 | 956 | 1178 | 1168 |

**Table 4.** Resultsobtained on generic TSP instances in terms of computational time.

| G | N | ACO | | ACOWU | | Dai | | Bellaachia | |
|---|---|---|---|---|---|---|---|---|---|
| | | Avg. | St.Dev. | Avg. | St.Dev. | Avg. | St.Dev. | Avg. | St.Dev. |
| 0 | 20 | 0.639 | 0.302 | 0.794 | 0.074 | 0.617 | 0.211 | 0.504 | 0.264 |
| 0 | 30 | 1.034 | 0.69 | 1.19 | 0.519 | 1.168 | 0.572 | 1.339 | 0.335 |
| 0 | 40 | 2.018 | 0.739 | 2.401 | 0.379 | 1.764 | 0.491 | 2.001 | 0.201 |
| 0 | 50 | 4.022 | 0.475 | **3.922** | 0.785 | 2.298 | 0.702 | 3.66 | 1.049 |
| 0 | 60 | 5.469 | 1.931 | **4.065** | 0.876 | 6.816 | 0.863 | 4.515 | 2.083 |
| 1 | 20 | 0.349 | 0.043 | 0.504 | 0.107 | 0.553 | 0.128 | 0.355 | 0.098 |
| 1 | 30 | 1.008 | 0.242 | 1.097 | 0.371 | 0.978 | 0.424 | 1.124 | 0.142 |
| 1 | 40 | 1.568 | 0.752 | 1.876 | 0.563 | 1.799 | 0.81 | 2.032 | 0.727 |
| 1 | 50 | 3.788 | 1.248 | **3.053** | 0.991 | 3.9 | 0.884 | 2.815 | 0.708 |
| 1 | 60 | 4.765 | 1.808 | 4.902 | 2.114 | 5.695 | 2.089 | 4.717 | 1.688 |
| 2 | 20 | 0.63 | 0.25 | **0.566** | 0.189 | 0.572 | 0.077 | 0.513 | 0.123 |
| 2 | 30 | 1.669 | 0.559 | **1.386** | 0.242 | 1.514 | 0.57 | 1.051 | 0.224 |
| 2 | 40 | 1.869 | 0.355 | 2.556 | 0.795 | 2.556 | 0.486 | 2.675 | 0.717 |
| 2 | 50 | 3.833 | 1.192 | 4.048 | 1.422 | 3.054 | 1.078 | 3.7 | 1.295 |
| 2 | 60 | 5.965 | 2.003 | 6.11 | 1.496 | 5.815 | 1.269 | 5.97 | 1.593 |
| 3 | 20 | 0.692 | 0.226 | 0.718 | 0.229 | 0.686 | 0.22 | 0.538 | 0.15 |
| 3 | 30 | 1.578 | 0.465 | **1.44** | 0.429 | 1.598 | 0.356 | 1.236 | 0.354 |
| 3 | 40 | 2.011 | 0.863 | 2.37 | 0.569 | 2.229 | 0.669 | 2.142 | 1.046 |
| 3 | 50 | 4.535 | 1.258 | **2.936** | 1.394 | 3.235 | 1.708 | 3.203 | 1.504 |
| 3 | 60 | 4.087 | 0.6 | 4.716 | 0.979 | 6.637 | 1.81 | 4.121 | 0.966 |
| 4 | 20 | 0.459 | 0.211 | 0.669 | 0.27 | 0.573 | 0.227 | 0.419 | 0.059 |
| 4 | 30 | 1.182 | 0.35 | 1.295 | 0.453 | 0.923 | 0.301 | 1.05 | 0.434 |
| 4 | 40 | 2.334 | 0.736 | **1.44** | 0.626 | 2.394 | 0.865 | 1.828 | 0.64 |
| 4 | 50 | 3.795 | 0.964 | **2.826** | 1.222 | 2.988 | 0.933 | 2.312 | 0.467 |
| 4 | 60 | 4.375 | 1.93 | **3.347** | 0.77 | 3.724 | 1.741 | 4.196 | 1.889 |

Tables 3 and 4, Figures 2 and 3 show the comparison in terms of computational time and solutions explored by the algorithms before finding the best one. In this sense, the Dai and Bellaachia algorithms are the best ones, although, the difference with the proposed ACOWU is not that big—i.e., 100–300 milliseconds, which translate into a few milliseconds. Moreover, even the ACOWU is able to outperform all the others in some experiments. The experiments in which the ACOWU needs less iterations (and consequently, computational time) to find the best solutions are also the most complicated instances where the number of nodes to visit is higher (i.e., 40–60 nodes).

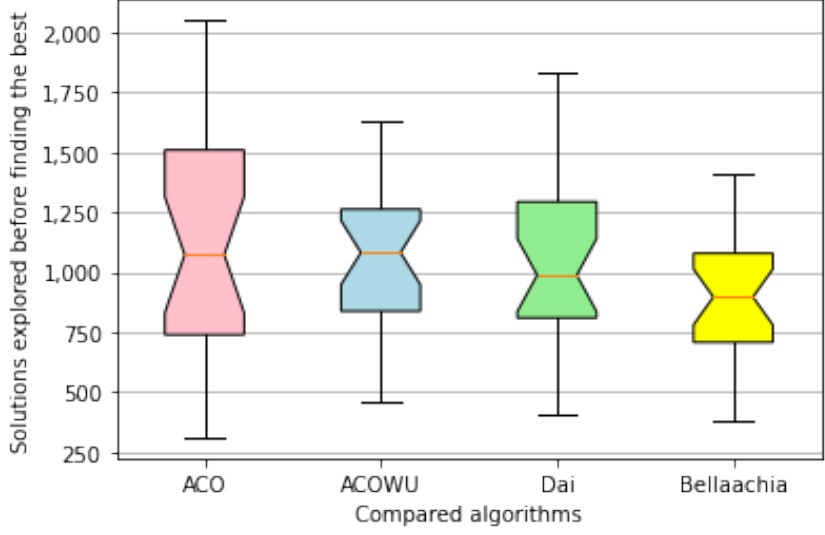

**Figure 2.** Comparison of iterations on generic TSP instances.

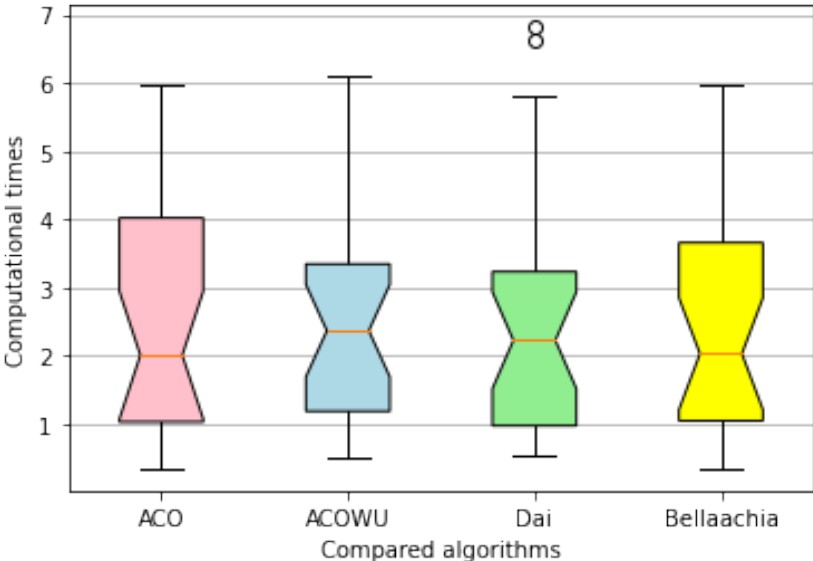

**Figure 3.** Comparison of solutions explored on generic TSP instances.

*4.5. Results Obtained in the Simulation of the Real Warehouse*

After studying the effect of the proposed warm-up on a set of generic TSP instances, it is in interest of the author to analyze its effect in a more realistic and complex environment. The simulation of a manual warehouse for picking was therefore used, and the proposed ant colony optimization with warm-up is used to define the routing of pickers—i.e., once defined the picking locations the picker has to visit, the order in which they are visited is defined. The faced problem is essentially a TSP, but the graph of nodes and paths is more constrained, with less possible paths between nodes and many mandatory walkways. Importantly, no additional constraints such as capacity of pickers' baskets, definition of batches, or interference between pickers moving through the aisles are considered.

Starting from the warehouse layout, a graph of accessible positions is generated placing a node in front of each storage location and a node where aisles cross to each other, and then, using the well-known Floyd-Warshall algorithm, the matrix of minimum distances between nodes is generated. The starting warehouse is made of 20 aisles with 16 storage locations each, crossed by a single cross-aisle in the middle (i.e., between the 8th and the 9th locations). Each storage location are 2×2 m, aisles are 4 m wide, while the cross-aisle is 8 m wide. The resulting graph used in the tests is shown in Figure 4.

The results obtained by the compared algorithms in the simulation of the warehouse are reported in Table 5 and can be intuitively visualised looking at Figures 5–7 respectively in terms of *(i)* cost of the best solution found, *(ii)* solutions explored before finding the best, and *(iii)* computational times. The results broadly respect what already seen in previous experiments. On average the proposed ACOWU is still the best in terms of cost even if sometimes it cannot provide a better solution than the classic ACO, but the same could be said for the other algorithms using a pheromone initialization strategy. The Dai algorithm is still in second position and proved to be a very good alternative. Concerning the solutions explored and therefore the computational time Bellaachia algorithm is the best (as already seen in previous experiments). However, the proposed ACOWU is again a good alternative as clearly visible in Figures 6 and 7. Again, as in the previous experiments on generic TSP instances, the difference in terms of solutions explored and computational time is not that big. However, the utilization of a pheromone initialization technique, as already proved in literature, guarantees some advantages over the classic ACO.

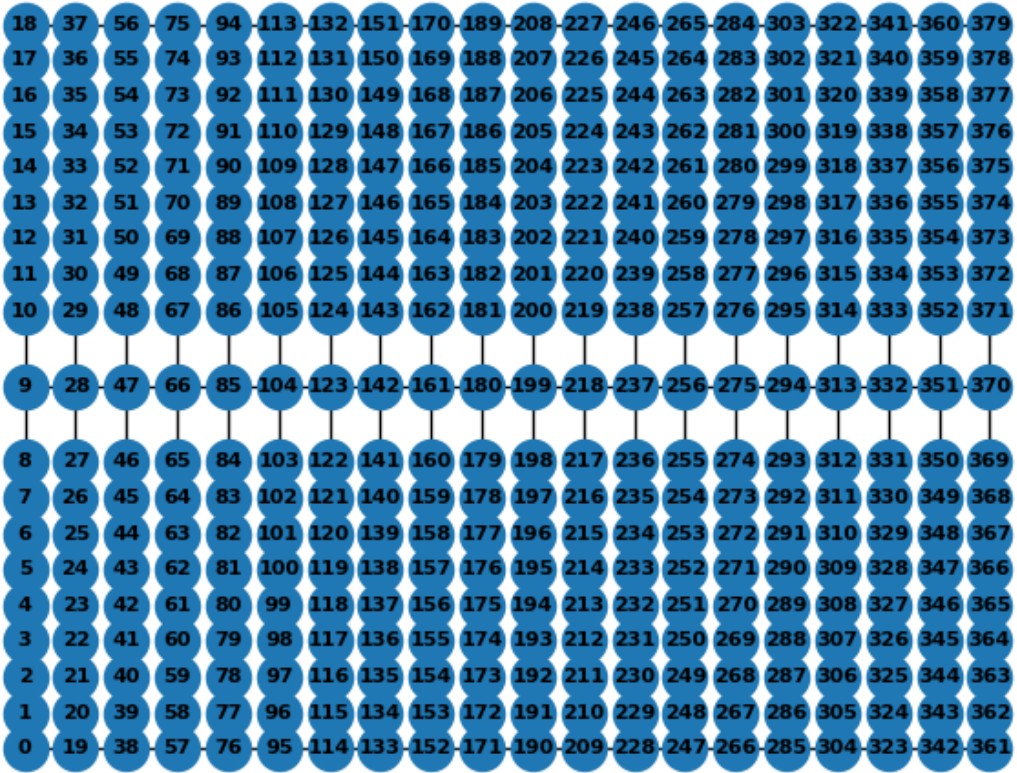

**Figure 4.** Graph of warehouse used for tests.

**Table 5.** Results obtained in warehouse.

| | Cost of the Best Solution Found | | | | | | | |
|---|---|---|---|---|---|---|---|---|
| | **ACO** | | **ACOWU** | | **Dai** | | **Bellaachia** | |
| **N** | **Avg.** | **St.Dev.** | **Avg.** | **St.Dev.** | **Avg.** | **St.Dev.** | **Avg.** | **St.Dev.** |
| 20 | 543 | 13 | 552 | 15 | 537 | 9 | 714 | 37 |
| 30 | 604 | 14 | 626 | 15 | 616 | 18 | 864 | 34 |
| 40 | 804 | 30 | **781** | 18 | 778 | 18 | 1070 | 44 |
| 50 | 806 | 30 | **755** | 8 | 792 | 43 | 1138 | 69 |
| 60 | 1026 | 144 | **1001** | 56 | 995 | 46 | 1420 | 74 |

| | Solutions Explored before Finding the Best | | | | | | | |
|---|---|---|---|---|---|---|---|---|
| | **ACO** | | **ACOWU** | | **Dai** | | **Bellaachia** | |
| **N** | **Avg.** | **St.Dev.** | **Avg.** | **St.Dev.** | **Avg.** | **St.Dev.** | **Avg.** | **St.Dev.** |
| 20 | 1157 | 738 | **707** | 520 | 610 | 555 | 335 | 246 |
| 30 | 1111 | 542 | **860** | 665 | 1091 | 293 | 921 | 477 |
| 40 | 1266 | 590 | **1041** | 343 | 1660 | 846 | 1230 | 280 |
| 50 | 1009 | 771 | 1721 | 781 | 1656 | 861 | 1096 | 466 |
| 60 | 1273 | 1036 | 1925 | 401 | 1460 | 950 | 544 | 373 |

| | Computational Times | | | | | | | |
|---|---|---|---|---|---|---|---|---|
| | **ACO** | | **ACOWU** | | **Dai** | | **Bellaachia** | |
| **N** | **Avg.** | **St.Dev.** | **Avg.** | **St.Dev.** | **Avg.** | **St.Dev.** | **Avg.** | **St.Dev.** |
| 20 | 0.473 | 0.162 | **0.366** | 0.111 | 0.365 | 0.121 | 0.289 | 0.055 |
| 30 | 0.982 | 0.249 | **0.872** | 0.309 | 0.948 | 0.131 | 0.876 | 0.218 |
| 40 | 1.824 | 0.43 | **1.617** | 0.266 | 1.961 | 0.494 | 1.722 | 0.217 |
| 50 | 2.334 | 0.838 | 3.103 | 0.748 | 3.022 | 0.716 | 2.494 | 0.553 |
| 60 | 3.821 | 1.626 | 4.847 | 0.425 | 3.983 | 1.274 | 2.672 | 0.653 |

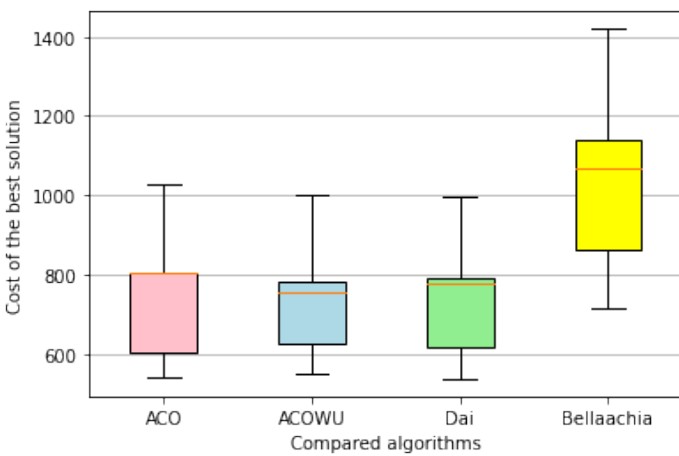

**Figure 5.** Results obtained in warehouse in terms of cost.

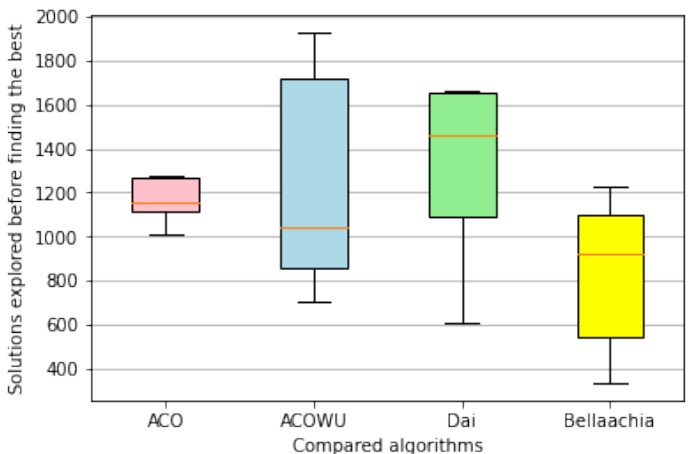

**Figure 6.** Results obtained in warehouse in terms of solutions explored.

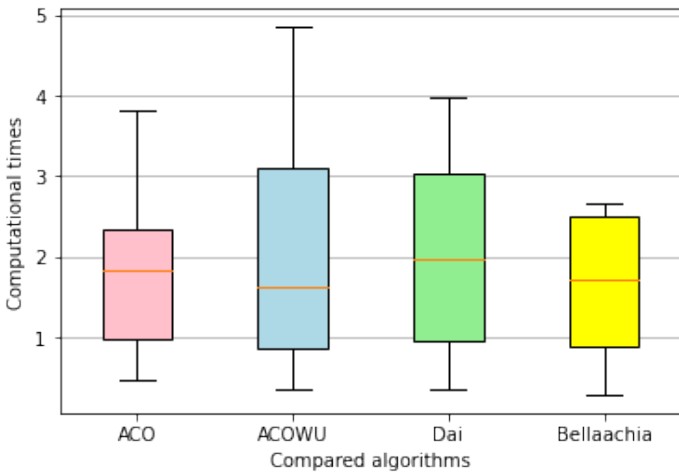

**Figure 7.** Results obtained in warehouse in terms of computational time.

## 5. Conclusions

In this paper, a warm-up procedure for the ACO was proposed and validated. During the warm-up, the pheromone matrix of the ACO is initialized to provide an efficient new starting point for the algorithm so that it can obtain the same (or better) results with less

iterations. The warm-up is based exclusively on the graph made by the nodes and the edges that formalize the problem. This graph, in most applications, is given, and does not need to be recalculated every time before executing the algorithm. Because of this, the warm-up procedure can be made only once when setting the hyper-parameters of the algorithm to speed it up every time it is used from then on. Firstly, a parameters tuning was made to find the optimal setting for the warm-up. Then, two set of the experiments were carried out to validate the proposed approach. The first set of experiments was done using some generic TSP instances, then, to validate algorithm in a more realistic context, a second set of experiments in a warehouse for picking was made. The ant colony with warm-up was compared with a classic ACO (without warm-up), and with two ACO using a pheromone initialization technique. The results obtained are promising, and the warm-up approach is generic enough to find application in almost all the contexts where the ACO can be applied. Of course, the impact and the efficiency of the warm-up might change from one application to the other, but the preliminary results shown in this paper prove that its analysis is worth studying, paving the way for many studies and possible extensions.

**Funding:** This research received no external funding.

**Data Availability Statement:** The code of the algorithm was made open-source the 1 September 2021 at https://github.com/mattianeroni/AntColony.

**Conflicts of Interest:** The author declares no conflict of interest.

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
