# Peer review of "Ant Colony Optimization with Warm-Up"

_algorithms, doi:10.3390/a14100295_

Round 1

Reviewer 1 Report

This paper presents the Ant Colony Optimization algorithm equipped with the warm-up procedure. The topic is interesting, and the paper is well-written. However, it requires some minor improvement based on the following points:

  1. The references should be entered in ascending order, starting from 1.
  2. The matrix of updates introduced at the bottom of (1) should be written italic.
  3. There are no references in line 124, where discussion related to parameters of ACO is shown.
  4. In the last paragraph p. 3.3. the author introduces information related to several versions of the ACO. References should support this discussion.
  5. It is advisable to provide the actual times necessary to solve optimization in both cases, i.e., with and without warm-up.

Author Response

I am very happy that the reviewer appreciated my work. The paper has been changed according to the suggestions provided. An accurate point-to-point response is reported below and all the changes to the text are highlighted in blue:

(1) The citations has been sorted in ascending order starting from 1 has suggested.

(2) The matrix of updates has been written in Italic.

(3) A reference has been included in line 124.

(4) Some references has been included at the end of section 3.3 in order to support my discussion about the parameters of ACO.

(5) The computational times relative to all the tests carried out has been included in the paper.

Reviewer 2 Report

The paper presents a warm up technique for accelerating the overall execution time of ACO algorithms. The paper is in my point of view well structured and presented.

The background points out that many variations exist about ACO family of metaheuristics. However, neither the backbround nor the experiments focus on the pheromone matrix initialisation step. I guess that work has been done about it. I suggest that for both related work and experiments, this has to be more accuratly explorated, e.g. comparisons must be done with either most well fitted ACO algorithms for TSP and/or other ACO algorithms with specific initialization steps instead of comparing with standard ACO and greedy heuristic.

typos: review numbers in outline L 57

Author Response

I would like to thank the reviewer for the time and effort he/she dedicated to my work, and I really appreciated he/she defined my work "well structured and presented". I believe the provided suggestions have been very helpful in order to improve the overall quality of the paper, and I therefore tried to modify the paper accordingly. 

According to the suggestion provided by the reviewer, two ACO that carry out a pheromone initialisation have been introduced, included in the paper, and compared with the proposed algorithm. Additional experiments have also been carried out using a real warehouse layout instead of generic instances of TSP.  According to the suggestions of the other reviewer, the computational times have been also recorder during the experiments and presented in the paper.

Round 2

Reviewer 1 Report

The Author addressed all my comments and remarks satisfactorily. Additional tests are interesting and stimulative to the readers. The paper can be accepted for publication. Congratulations.